# Striking Variability in the Post-Reproductive Movements of Spanish Red Kites (*Milvus milvus*): Three Strategies, Sex Differences, and Changes over Time

**DOI:** 10.3390/ani12212930

**Published:** 2022-10-25

**Authors:** Jorge García-Macía, Andrea Pomares, Javier De la Puente, Ana Bermejo, Juan Martínez, Ernesto Álvarez, Sara Morollón, Vicente Urios

**Affiliations:** 1Vertebrates Zoology Research Group, University of Alicante, Apdo. 99, E-03080 Alicante, Spain; 2SEO/BirdLife, Bird Monitoring Unit, C/Melquiades Biencinto, 34, E-28053 Madrid, Spain; 3GREFA (Grupo de Rehabilitación de la Fauna Autóctona y su Hábitat), C/Monte del Pilar S/N, E-28220 Majadahonda, Spain

**Keywords:** raptors, spatial ecology, post-breeding, dispersive migration, sedentarism with post-reproductive movements

## Abstract

**Simple Summary:**

It is necessary to study the variability of movements in birds during their entire lifecycles. Many species and populations are classified into sedentary or migratory, but our results suggested that the Spanish breeding red kites have a wider range of spatial strategies: sedentarism, intra-peninsular migration and sedentarism with post-reproductive movements (or dispersive migration). This last spatial strategy was much more common in females. Furthermore, we reported year-to-year variability, so individuals could shift from one strategy to another. These results reveals for the first time the variability of movements in the Spanish red kites.

**Abstract:**

It was assumed that the Spanish breeding population of the red kite (*Milvus milvus*) was resident, hence their movements were restricted to their breeding area for their entire lifecycle. However, recent observations indicated that the post-reproductive strategies of the red kite in Spain are more diverse. We tagged 47 breeding adult red kites in Spain and analyzed their movements during the post-reproductive period (July–February). We found three strategies in the population: migration (10%), sedentarism (70%), and sedentarism with post-reproductive movements (20%), based on seasonality and other movement parameters. Sedentarism with post-reproductive movements was a very variable strategy that involved all-direction wandering movements far away from the nest (up to 589 km) after breeding season, and then a returned journey toward the starting point in time for the next breeding season. Our results also suggest that sedentarism with post-reproductive movements is much more common in females than males. Furthermore, 17% of the individuals changed their strategy over the years. This study highlights the great individual variability and plasticity of the red kite and allows for a better understanding of spatial ecology in opportunistic raptors.

## 1. Introduction

It is usually convenient to classify birds as sedentary or migratory. Sedentarism consists of remaining in the same area for the entire year. Sedentary birds perform short movements, in all directions, restricted to a circumscribed home range [1,2,3]. Contrarily, migratory birds use two main areas throughout the year: breeding and wintering areas. Migrating individuals make regular and fixed movements between those two areas twice a year (spring and autumn), involving long journeys of tens, hundreds, or thousands of kilometers [4,5,6]. In the northern hemisphere, migration allows birds to be in higher latitudes during the summer, and in lower latitudes during the winter, which entails some environmental benefits. In this sense, migration often has a very marked latitudinal component [7,8,9,10,11], but longitudinal migration also exists [12].

However, this dual classification may be insufficient for the actual diversity of movement patterns that can be found in birds. For example, Newton [13] suggested six types of birds’ movements according to different criteria like direction, distance, or calendar dates: routine day-to-day movements, dispersal movements, dispersive migration, migration, irruptions, and nomadism. 

Dispersive migration concerns post-breeding movements made by individuals commonly considered “residents”. These birds make long movements from their breeding areas, without marked seasonality or direction, followed by a return journey. This type of post-reproductive movement has been previously identified, under various terms, in some typically sedentary raptors such as the golden eagle (*Aquila chrysaetos*) [14], the bald eagle (*Haliaeetus leucocephalus*) [15], the eagle owl (*Bubo bubo*) [16], or the burrowing owl (*Athene cunicularia*) [17]. These species leave their reproductive territories after breeding to explore areas far away from them. Dispersive migration has also been observed in typically migratory raptors such as the western marsh harrier (*Circus aeruginosus*) [18], or the Montagu’s harrier (*Circus pygargus*) [19], during pre-migratory or post-migratory periods. In short, dispersive migration is very similar to the wandering dispersal movements of immatures but performed by adults after breeding. Although shortages of food could be one of the drivers of this strategy [20,21], little is yet known about why, how, and when birds perform dispersive migration.

The red kite (*Milvus milvus* Linnaeus 1758, Accipitridae) is a medium-sized raptor with a western Palearctic distribution: found in Europe, North Africa, and some Atlantic islands [22]. This species has great variability in its spatial strategies. The largest number of reproductive pairs from Central Europe perform an intra-continental migration to their wintering areas located in southern Europe [23,24,25,26,27,28,29]. However, there are also resident breeding populations in different European regions, especially in the southern limits of its distribution [29,30]), including more than 2000 pairs in Spain [31]. The relevance of these populations may increase in the future because a sedentarization process is taking place in this species [24,32], and many others [33,34,35,36]. Global changes, including changes in land use, rising temperatures, and the loss of habitats, may be one of the drivers of the progressive change in the behavior of the species, which must face new threats in highly changing environments [24,37,38].

Until now, it has been assumed that the Spanish breeding population of the red kite was strictly resident. However, recent observations may indicate that the diversity of strategies of this population is much higher. In this study, the movements of 47 adult red kites after their breeding season in Spain were studied using GPS telemetry. The objectives of this study were to (1) define and analyze the post-reproductive movements from July to February using parameters such as latitudinal displacement, nights outside the breeding area, maximum distance to nest, and mean distance to nest; (2) classify the post-breeding periods based on the strategy (sedentary, sedentary with post-reproductive movements, or migratory), and test the differences between them; (3) study the year-on-year shifts in the strategies of the individuals; and (4) analyze sex differences.

## 2. Methods

### 2.1. Study area and Tagging

Forty-seven breeding adult red kites were tagged in Spain from 2013 to 2020 (34 females, 12 males, and one individual with an undetermined sex), providing data for 136 post-breeding periods (Appendix A). They were captured in different Spanish provinces (Appendix A): Madrid (17), Soria (5), Álava (4), Guipúzcoa (4), Ávila (3), Cáceres (2), Salamanca (2), Segovia (2), Burgos (1), Huesca (1), León (1), Palencia (1), Toledo (1), Valladolid (1), Zamora (1), and Zaragoza (1).

The individuals were, at least, three years old in their first period considered (Euring codes 7 and 8). All individuals reproduced successfully in Spain and did not leave the Iberian Peninsula for their entire lifecycle. The breeding season in the red kite lasts from March to June [39], so we analyzed the post-reproductive period, that is, from July to February, both included. 

The individuals were captured using a dho-gaza net with a decoy of a live eagle owl (*Bubo bubo*), close to active nests with chicks at least 15 days old [40]. Some individuals were tagged as chicks in their nests and became breeding adults. All individuals were weighed, measured, and ringed, and a blood sample was taken for molecular sexing [41]. A GPS transmitter was attached to the back of the individuals using a Teflon harness, a non-abrasive material tied with cotton thread to the body of the animal. Three transmitter models were used: the 20 to 23 g SAKER, DUCK or SKUA GPS-GSM (Ecotone Telemetry, Gdynia, Poland; n = 37), the 22-g PTT-100 solar-powered Argos/GPS (Microwave Telemetry Inc., Columbia, MD, USA; n = 3), and the 20-25-30g OrniTrack-20, OrniTrack-25 and OrniTrack-30 solar powered GPS-GSM tracker (Ornitela, Vilnius, Lithuania; n = 7).

All animal handling and marking were approved by an ethical committee, and carried out under the supervision of licensed banders from the Spanish Society of Ornithology (SEO/BirdLife) and GREFA (Grupo de Rehabilitación de Fauna Autóctona y su Hábitat). All the individuals were captured and tagged with the prior permission of local administrations (https://www.ae.jcyl.es/verDocumentos/ver?loun=COM51WDX6O4G57M02DGBL1 (accessed on 20 October 2022)), in accordance with the national Spanish Law 42/2007 of 13 December 2007 on Natural Heritage and Biodiversity. The weight and characteristics of the transmitters were in accordance with scientific standards, so no significant behavioral differences were found in the individuals due to the transmitters.

The transmitters were programmed with different frequencies depending on their capacity and the hour of day. The Ecotone transmitterswere programmed to obtain locations every hour from 06:00 to 19:00, the Microwave provided a location every two hours from 06:00 to 18:00, and the Ornitela provided locations every five minutes from sunrise to dusk. The differences regarding the frequency of the emission of locations by the different transmitters did not substantially affect the results given the variables selected for the work. 

### 2.2. Spatial Parameters

Based on recent studies on raptor movements [25,39,42], four variables were considered to analyze the movements of each period: maximum distance from the nest (km), that is, the Euclidean distance of the farthest location from the nest; the mean Euclidean distance from the nest (km) using daily positions; the maximum latitudinal displacement (°) from the nest; and the nights spent outside the breeding area (>50 km, based on own data from the breeding season of the individuals). For the calculation of the mean distance to the nest, values were taken close to midnight to ensure that the individual was resting. For periods without midnight locations, values were chosen in the early morning or as early as possible, always after 6:00 p.m. and before 8:00 a.m. Values with more than 24 h difference were discarded. 

### 2.3. Strategies Classification

Prior to the analyses, we performed a qualitative classification of the post-reproductive periods of the individuals. We delimited three categories (sedentarism, sedentarism with post-reproductive movements, and migration) based on two characteristics (Figure 1): first, the presence or absence of long movements away from the nest (>50 km, based on own data of the individuals during the breeding period, which did not move beyond that distance); second, if they spent the entire wintering season without returning to the breeding area.

This classification was based on previous studies and reviews [13], and these strategies can be described as (Figure 2):-Migration. Individuals that performed two clear latitudinal movements in the post-reproductive period, one in autumn, from their nest to their wintering areas, and another in spring, from their wintering areas back to the nest. These individuals clearly differed from the others because they divided their year into these two areas and made movements between them with the indicated periodicity.-Sedentarism. Individuals that remained very close to the nest for the entire year, without significant displacements. They only performed routine day-to-day movements throughout the year.-Sedentarism with post-reproductive movements (or dispersive migration). Individuals which performed all-direction movements far away from their nests after breeding season, and then a return movement toward the starting point in time for the next breeding season. It involved both wandering movements and settling in temporary settlement areas. These birds were not classified as migrators because their movements had no marked periodicity, since they could perform from one to several movements after breeding, with different durations and distances. Moreover, they did not prefer a specific wintering area, as with migrators. They were also clearly distinguishable from the sedentary because they performed clear displacements from the nest of a considerable distance. These movements have a component more exploratory and dispersive, as is shown in previous works [14,15,16,17], similarly to the dispersal movements of immature birds (<2 years old).

### 2.4. Statistical Analyses

Once periods were quantitatively classified, Principal Component Analysis (PCA), calculated on a correlation matrix (Appendix A), was used to explain the differences between the three categories. Additionally, four different LMMs (Linear Mixed Models) were performed to check the differences between the different strategies. In all models, “strategy” was considered as a “fixed factor”, while “individual” and “period” were considered as “random factors”, and “period” was nested into “year” to account for the non-independence of the data. The response variables were the “maximum latitudinal distance”, the “number of days outside the breeding area”, the “maximum distance to the nest”, and the “mean distance to the nest”. In order to evaluate the significance in linear mixed-effects, ANOVA tests with Kenward-Roger approximations were performed. Finally, pairwise comparisons using Student’s t-tests with a pooled SD were used [43]. Models were computed using the package “lme4” for R [44].

Fisher’s exact test (data were non-parametric) was used to check differences in the distribution of the strategies between males and females. 

All statistical analyses were performed using R software version 4.0.3 (R Foundation for Statistical Computing, Vienna, Austria) [45], and the significance level was established at *p* < 0.05. QGIS software version 3.12 (QGIS Association, Switzerland) was used to visualize the movements of the individuals and make the maps.

## 3. Results

### 3.1. Strategies Classification

All post-reproductive periods (July–February) were classified into three strategies: migration, sedentarism, and sedentarism with post-reproductive movements (Figure 3). Sedentarism was the most common strategy, followed by sedentarism with post-reproductive movements, and migration in last place (Appendix A). In total, 59 sedentary (65%), 25 sedentary with post-reproductive movements (30%), and 7 migratory (8%) periods were found in females (n = 91 periods in 34 individuals). Moreover, 35 sedentary (82%), 2 sedentary with post-reproductive movements (4%), and 6 migratory (14%) periods were found in males (n = 43 periods in 14 individual).

### 3.2. Spatial Parameters

Sedentary periods were characterized by lower latitudinal displacement, maximum distance reached, and mean distance to the nest. On the other hand, migration periods showed the higher values of these variables. Sedentary with post-reproductive period values were in the middle between those two strategies (Table 1, Figure 4). Despite the high individual variability, there were differences between the three strategies for all movement variables (Table 2 and Appendix A; *p* < 0.0001 for all pairwise comparisons). 

Sedentarism with post-reproductive movements was the most diverse strategy, showing great variability in relation to the dates, number, duration, and distances of those movements. The movements were more frequent just after breeding season, but several individuals performed them for the entire year until the next breeding period. The red kites nesting in the North of the Iberian Peninsula usually made these movements toward the South, and those with a nest in the South usually made movements toward the North (Appendix A). In contrast, individuals that nest in the center of the Iberian Peninsula moved in all directions. Some individuals only performed one movement that lasted just for a few days, while others (performed several movements of different durations. During these periods many individuals combined the permanence in low-mobility temporary areas with wandering movements throughout the territory. All individuals repeated the same breeding area in the following years.

There were sex differences in the distribution of individuals among the different strategies (*p* < 0.001). The percentage of sedentary and migratory periods was similar in males and females, but the percentage of sedentary with post-reproductive periods was significantly higher in females (30%) than males (4%).

### 3.3. Year-on-Year Variability of Strategies

In total, 9 of the 47 tagged red kites changed their strategy between years, at least once (Table 3 and Appendix A). Two males (16.7% of total males) changed their strategy from sedentary to migratory, or vice versa. Seven females (17.6%) also changed their strategy: four of them changed from sedentary to sedentary with post-reproductive movements or vice versa, and two changed from migration to sedentary with post-reproductive movements.

For example, Madrid01 (male) performed a migratory strategy for its first four years, and then it became sedentary for the next three years. Toledo02 (male) migrated in its first year, and then it became sedentary with post-reproductive movements and sedentary for two years, and migrated again in its fourth year. Soria05 (female) migrated during its first year, before becoming sedentary with post-reproductive movements for all the following years. Zaragoza02 (female) was sedentary with post-reproductive movements during its first four years, but it became sedentary after the fifth year (Appendix A).

## 4. Discussion

In this work, the post-reproductive spatial strategies of the Spanish red kites were studied. Until now it has been assumed that Spanish breeding red kites were sedentary, but here it was supported that a small part of the individuals migrates within the Iberian Peninsula, and another part, mainly females, performs wandering post-reproductive movements. Moreover, 17% of the red kites changed their strategy throughout their lifecycle. 

Despite the high individual variability and the frequent problems with classifying the strategies of birds, at least three differentiated strategies in the Iberian breeding red kites could be established: migration, sedentarism, and sedentarism with post-reproductive movements.

Intra-peninsular migrators represented the lower percentage of Spanish breeding red kites (10%), which performed short-distance travels across the Iberian Peninsula. Red kites’ migration from central Europe to southern Europe is well documented [25,30], but this shorter migration within the Iberian Peninsula was poorly known. Spain is a relatively small territory on the southern limit of the red kite distribution, commonly used by this species for wintering, so it was expected that migration was not necessary for breeding individuals in most cases. Even so, there were some individuals which took advantage of the high environmental differences between the North and South of Spain and found optimal conditions to spend the winter far away from their breeding areas [46,47]. Other species, like the booted eagle (*Aquila pennata*), exhibit similar behavior: most of the individuals migrate to Africa, but some of them spend the winter in the South of the Iberian Peninsula [33,48,49].

There were differences between male and female red kites in the post-breeding period. Migration and sedentarism were equally frequent in both sexes, but the sedentarism with post-reproductive movements strategy was much more frequent in females than in males. Data about the pairs were not available in this study, but the sex differences may be explained by two factors. First, if females left the nest during the post-breeding period, they would avoid intra-specific competition in the breeding area at a time of fewer food resources [50]. Second, the permanence of the males near the nest throughout the entire year could be an effective way to protect and maintain the territory until the next breeding period. Female red kites from Central Europe usually travel further distances during dispersal and migration than males [28]. Similarly, our results support that Spanish female red kites have greater mobility in the post-reproductive period. 

Temporary settlement areas were often used by the individuals during sedentarism with post-reproductive periods. Individuals may select these areas because of their food abundance and prey richness [39]. Given that these areas are used by several individuals throughout the year, or even by the same individual during the same year, they should be prioritized for the conservation of the species, contributing to the delimitation of SPAs (Special Protection Areas) and other protected zones. In these areas, in addition to a significant accumulation of red kites, other raptors could be present at this time for the same reasons [51,52]. The protection of these temporary settlement areas could decrease the high mortality of the red kite in its dispersive or post-breeding phases and ensure the reproduction of females in the next year. In this way, the conservation strategy of the species should not only be extended to nesting places but should also consider their post-reproductive movements. A more exhaustive study that delimits and characterizes these areas for the population as a whole is needed. 

In total, 17% of the individuals changed their post-reproductive strategy over the years. Strategies seem to be selected regardless of the latitude of nests and age. The drivers of the selection of one strategy or another could not be identified based on our results but may be linked to a set of inter-related factors. Some studies demonstrated that birds could quickly change their strategy (from sedentary to migratory, or vice versa) due to selective pressure [53,54,55]. In regions with a mild climate, a great availability of food, and few predators or competitors, sedentary behavior is favored. On the contrary, a lack of food or increased seasonality favors migration [5,55,56,57]. The shifts between the three described strategies in the red kite may respond to the same reasons, so individuals select one or another depending on weather, the variability of food, the degree of seasonality, breeding success, productivity, etc.

The high plasticity of the red kite may be a positive factor for maintaining stable populations, but this has not happened in recent decades, especially in the Spanish breeding population [31,32]. The high mobility of the species may have some disadvantages. On the one hand, it might increase mortality due to the lack of information on the territory and travel routes, which could cause collisions, electrocutions, etc. On the other hand, red kites usually feed in landfills and middens [58] during wandering movements, even settling there, which may lead to an increase in mortality.

## 5. Conclusions

The Spanish breeding population of the red kite showed great spatial plasticity: most of the individuals are resident, but there are also intra-peninsular migrators and a third diverse group that performs wandering movements combined with settling in temporary areas far away from the nest, which may be called “sedentarism with post-reproductive movements”. Sedentarism with post-reproductive movements was much more frequent in females than in males. Furthermore, 17% of the individuals changed from one strategy to another over the years. This study provided several findings that are key to understanding the behavioral plasticity of the red kite and other opportunistic raptors, which could help with the conservation of these species.

## Figures and Tables

**Figure 1 animals-12-02930-f001:**
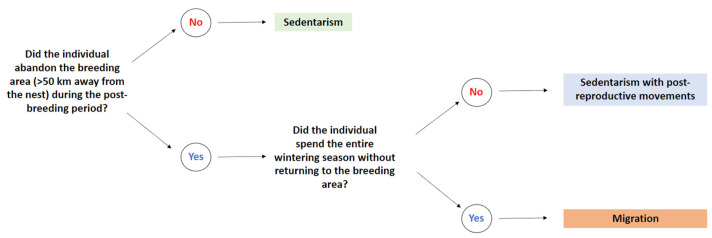
Qualitative classification criteria to delimit the three post-reproductive strategies.

**Figure 2 animals-12-02930-f002:**
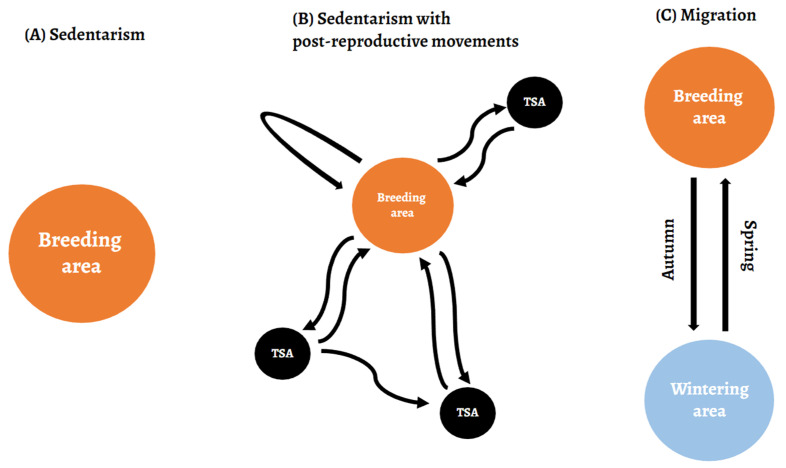
Scheme of the strategies followed by the red kites. (**A**) Sedentarism: routine day-to-day movements restricted to a circumscribed home range; (**B**) Sedentarism with post-reproductive movements: a combination of wandering movements and stopping in temporary settlement areas (TSA) after breeding season, without marked seasonality or direction, involving a return journey after that in time to next breeding season; (**C**) Migration: regular and fixed movements between two areas twice a year (spring and autumn).

**Figure 3 animals-12-02930-f003:**
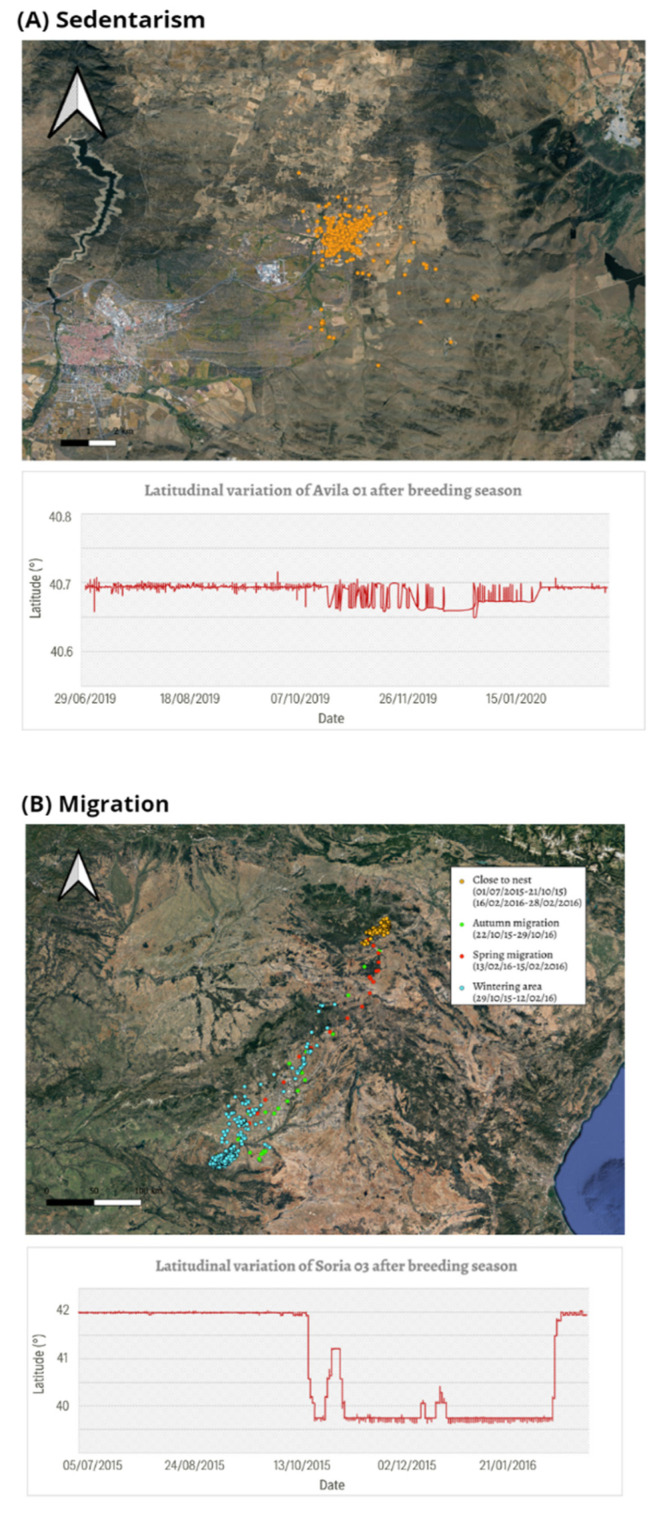
Examples of the different strategies performed by the Spanish red kites: sedentarism (**A**), migration (**B**), and sedentarism with post-reproductive periods (**C**). Each map represents the movements of one individual during a single post-breeding period. The latitudinal variation of each individual (Avila01, Soria03, and Palencia01, respectively) is represented below the map.

**Figure 4 animals-12-02930-f004:**
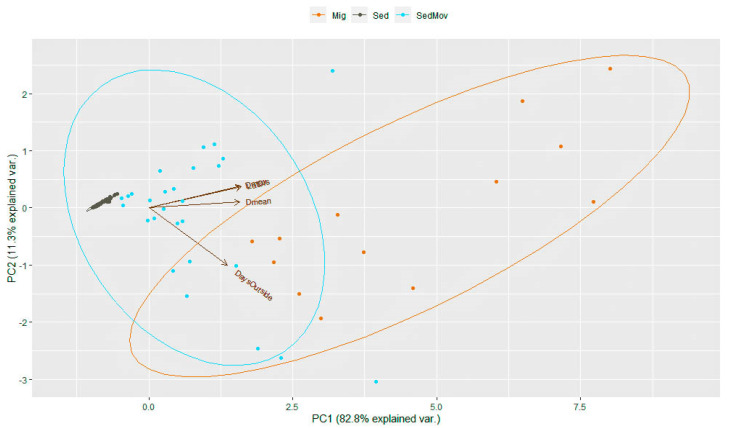
Principal component analysis (PCA) based on movement parameters (days outside breeding area, mean distance to nest, maximum distance to nest, and maximum latitudinal displacement. As depicted, 80% probability ellipsoids encompass the three strategies (orange = migratory; grey = sedentary; blue = sedentary with post-reproductive movements).

**Table 1 animals-12-02930-t001:** Movement parameters (maximum distance to nest, mean distance to nest, and daily traveled distance) broken down by strategies (sedentarism, sedentary with post-reproductive movements, and migration). The sample unit was the post-reproductive period (n = 136), so one individual with several years can be found in different strategies. Values are expressed as mean ± standard deviation (minimum–maximum).

	Number of Post-Reproductive Periods (%)	Maximum Latitudinal Variation from the Nest (°)	Nights outside Breeding Area	Maximum Distance to Nest (km)	Mean Distance to Nest (km)
Sedentarism	95 (70%)	0.2 ± 0.1 (0.1–0.5)	0 ± 0	24 ± 13 (3–49)	4 ± 3 (1–17)
Sedentarism with post-reproductive movements	28 (20%)	1.1 ± 0.8 (0.1–4)	44 ± 55 (3–216)	180 ± 115 (52–589)	23 ± 19 (5–98)
Migration	13 (10%)	2.8 ± 1.8 (1–5.3)	110 ± 32 (64–162)	435 ± 195 (249–781)	168 ± 89 (51–372)

**Table 2 animals-12-02930-t002:** Results of the ANOVA tests based on the previous LMMs (see Appendix A). Estimates, F value, degrees of freedom (df), degrees of freedom of the residuals (df.res), and *p* value are shown. n = 136. *p* < 0.0001 for all strategy pairwise comparisons.

Variable	Factor	F	df	df.res	*p*
Latitudinal displacement (°)	Intercept	136.81	1	53.408	<0.0001
	Strategy	74.85	2	125.13	<0.0001
Days outside breeding area	Intercept	136.06	1	51.303	<0.0001
	Strategy	90.12	2	128.97	<0.0001
Mean distance to nest (km)	Intercept	301.95	1	25.58	<0.0001
	Strategy	165.26	2	103.32	<0.0001
Maximum distance to nest (km)	Intercept	278.21	1	26.97	<0.0001
	Strategy	129.35	2	115.26	<0.0001

**Table 3 animals-12-02930-t003:** Strategy changes in the individuals between post-reproductive periods. “↔” indicates at least one change in the individual from one strategy to another, or vice versa.

	Male	Female
Number of individuals	12	34
Number of individuals with changes of strategy	2 (16.7%)	6 (17.6%)
Sedentarism ↔ Migration	2	0
Sedentarism ↔ Sedentary with post-reproductive movements	0	4
Migration ↔ Sedentary with post-reproductive movements	0	2

## Data Availability

GPS locations of tagged individuals are available at movebank.org under the authors permission.

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
