# Peer review of "Striking Variability in the Post-Reproductive Movements of Spanish Red Kites (Milvus milvus): Three Strategies, Sex Differences, and Changes over Time"

_animals, 2022, doi:10.3390/ani12212930_

Round 1

Reviewer 1 Report

Review: Striking variability in the post-reproductive movements of Spanish Red Kites (Milvus mivus): three strategies, sex differences, and changes over time

This is a nicely written manuscript describing the movement behavior of adult kites (n=47) of each gender following breeding. The authors used GPS data to study the maximum and average distance of kites to their nest over time, as well as the daily distanced travelled. During the period between July to February, they found the majority of kites (65.9%) to stay closely to the nest (mean distance 3.9 km), which they defined as sedentary individuals. In contrast to these individuals, they found about 10% of the kites (n=14) to express a clear seasonal migratory movement (mean distance 161 km, staying in a seasonal area). Next to these two distinct groups, they also found some female kites (no males) to make larger movements of varying length (both in total distance and number of days away) from the nest, which they defined as sedentarism with post-reproductive movements. The authors also found quite some flexibility among individuals following a given strategy in a given year

 I like the manuscript and I think it provides important knowledge for the conservation of Spanish Red Kites relevant to highlight species plasticity. Yet, I think the authors could include more information in their analyses (e.g., number of days a being ‘away’) and need consider using another approach to identify differences among movements (e.g., PCA, https://www.r-bloggers.com/2021/05/principal-component-analysis-pca-in-r/). Most importantly, I think the authors should apply a more objective way to model and identify movement strategies. Currently, the authors based their analysis on a linear mixed model where they apply predefined strategies (added as a fixed factor). Given the information provided in Table 2, the individuals expressing movements linked to ‘sedentarism’ and ‘sedentarism with post-reproductive movements’ have a considerable overlap in all parameters addressed (i.e. max, mean distances, and distance travelled per day). It is not clear to me what makes them classifying an individual that had travelled for example a maximum distance of 39.9 km as ‘sedentarism with post-reproductive movements’ whereas an individual that moved 86 km is classified as ‘sedentary’. For me, it looks more like the same strategy where some female individuals just have some larger home ranges by utilizing resources that are further away. Please clarify why/how you decided for your classifications. However, most importantly, to identify differences among expressed movements, strategies should not be pre-defined (see references listed below). In addition, to test for differences between ‘sedentary’ and ‘sedentarism with post-reproductive movements’, the linear mixed model should have ‘sedentary’ in the intercept instead of the migratory movement, alternatively using ‘contrasts’ to test for differences between each group. Currently, the statistical results show the how ‘migratory’ differed to the two others, but not in-between SedMov and Sed. I also strongly recommend moving Table S2 in the main document as it provides key knowledge for the entire result section  

Specific comments

Methods – please add a statement that all animal handling and marking has been approved by an ethical committee and add the document IDs.

Line 96: incomplete sentence?

Line 149-153: Net square displacement (NSD) is commonly applied measure to discriminate objectively among movement strategies, including the time being ‘away’. Please clarify why you decided on considering the max and mean distances only. Please have a look at the references below.

https://academic.oup.com/book/27547/chapter-abstract/197542295?redirectedFrom=fulltext; https://journals.plos.org/plosone/article?id=10.1371/journal.pone.0149594

https://onlinelibrary.wiley.com/doi/abs/10.1111/ecog.02587

https://besjournals.onlinelibrary.wiley.com/doi/10.1111/j.1365-2656.2010.01776.x

Line 151: please rephrase to: …the Euclidean distance of the farthest location from the nest; the mean Euclidean distance from the nest (km) using daily positions

Line 276: I am confused about the term ‘short’-migrators. Please clarify what you mean with short-migrators. According to Table 2 these individuals moved the largest max and mean distance to the nest and traveled the longest daily distance…

Section 3.2 please clarify the number of individual female kites showing these movements. It would be also be important to know whether an individual female showing these movements in following years, indeed is utilizing the same area. Looking in Table S1, this might indeed be the case at least for some of the females.

Figure 4: please increase font size and contrast - difficult to read

Table 1. please develop the table text with more information. For example,

‘Classification of post-reproductive periods (July-February) by sex and strategy in Spanish Red Kites (n=47, equals to 136 kites years), 2013-2020. Based on distances travelled to the next, movements were classified as sedentarism, sedentarism with post-reproductive movements and migration. ‘

Table 2. please develop the table text to clarify that one animal can be found in different strategies

Table S2 provides key information and thus should be placed in the main document. I recommend moving the current Table 1 into the Appendix instead

Reviewer 2 Report

This is a very interesting study that shows the change in the migration pattern of the red kite in recent years, which is probably related to climate change. This study includes a good sample size considering that it works with protected wildlife. Throughout the manuscript, we can see the effort in the coordination and logistics of different teams to obtain the necessary data for the study.

However, the manuscript is very poor despite having some really interesting data. ¡Be careful! When writing a scientific article, the first person (singular or plural) is never used. It is always written in the passive.

The introduction has a good structure and is well developed, although some references could use updating (for example, the red kite distribution by Cramps and Simmons, 1979; I'm sure there is some more recent study).

The methodology is poor and very messy. The first thing would be to describe the capture and handling of the animals, and how the age is calculated (by biometry? by feather development?). Also if there is anything else that researchers did with animals, such as a veterinary exam (somewhere you specified that all the animals were healthy, but you did not mention that a clinical exam was performed). Next, describe the different types of transmitters and finally comment that data were collected from the post-reproductive period of adults older than 3 years. It would be interesting to make a map with the points where the captures were made.

The figures of the results are really visual and help to understand the three strategies very easily (beware, figure 5 has been moved to the end of the article). However, the statistics seem to be wrong. If the confidence interval is higher than the mean, it does not make much sense to give that data (see table 2). This can happen for two reasons: first, because the data are very dispersed from each other, so instead of giving means it would be more useful to give the modes; the second because there is one or two data that is at one end, so I recommend removing that data from the statistics so that they do not alter it. Please review carefully the statistical analysis with an expert. As for the tables, the data contained in the tables should not appear in the text and vice versa. That is, no information is repeated. In this sense, table 1 should be eliminated.

On the other hand, when submitting a manuscript to a journal it is important to carefully review the format. Throughout the text, there are double spaces, merged words, the absence of punctuation marks, changes in font, etc. But perhaps one of the most serious faults is that the references are not in the format indicated by the journal. The tables are also not in the format indicated by the journal. Please, I suggest to authors that before submitting an article to any journal, they review the guidelines published on the site to ensure that everything is as it should be.

Finally, it is noted that the authors are not fluent in English. The translation of some sentences is quite bad, so I strongly recommend a language review. A couple of important notes: the cardinal points in English are written in capital letters, and the species of animals are in lower case unless they contain a proper name or a geographical region (eg Bonelli's eagle). Please check this in the text as they are constant bugs throughout the manuscript.

Despite the flaws, I consider that the data collected by the authors are of great interest to the scientific community, which is why I believe that it is necessary to improve the text for reconsideration.

Round 2

Reviewer 1 Report

Review revised version: Striking variability in the post-reproductive movements of 2 Spanish red kites (Milvus milvus): three strategies, sex 3 differences, and changes over time

I thank the authors for the effort they have put into the revision and in addressing my comments. I think the revised manuscript has improved and I appreciate seeing the PCA analysis, which I think adds important information to the manuscript.

However, I still have some (minor) comments:

Table 2: I still think it would be more logical having the sedentary  strategy in the intercept. I am missing the results of the post hoc test, you mentioned in your item-to-item response. Please clarify, and please excuse if I have missed something.

Fig 3-5: you want to consider to merge these figures into one, having all strategies next to each other (in the landscape view) and marked as 1a), 2a), 3a), 1b), and so on. Would be neat

Fig 6: I suggest moving this figure to the appendix. You have already many figures in your manuscript, and I think this is less relevant for the overall message of your study.  

Fig 7: Nice! But please avoid using red and green in the same figures – some people are color-blind and will not be able to distinguish these two colors

Reviewer 2 Report

The authors have modified the text enough to significantly improve the scientific quality of the text. They have included new statistical analyses, completed the description of the methodology and updated some references as requested.

However, there are still flaws in the format. When the review was sent to them commenting on how important it is to have OK the format, the authors responded as follows:

It is true that many formatting issues were not perfectly finished after uploading the first manuscript and the citation system do not fit with the journal style, but we decided to send it this way because MDPI says in the Instructions for Authors (https://www. .mdpi.com/journal/animals/instructions) the following: "We do not have strict formatting requirements [...] When your manuscript reaches the revision stage, you will be requested to format the manuscript according to the journal guidelines." We hope to resolve all these issues in this manuscript version.

When submitting a manuscript to a publisher, a project to an institution, or a even CV to a company, it is expected that the text is carefully written. The revision that we carry out is at a scientific level, not grammatical. Minor language corrections may arise as not all of us have a good level of English. But presenting an article with changes in font size, justified and unjustified paragraphs, double spaces, merged words, and misspellings take professionalism off to the authors. Even more when they had already been warned about this situation and they not have carefully reviewed the text and they hide behind the MDPI guidelines. When in the MDPI guides it indicates that there is no strict format, it does not refer to this: 

The tables have a much larger font than the manuscript, as does the reference section, which shouldn't be the case.

Some paragraphs are misaligned with text and figures out of place (you can't start a section directly with a figure).

The title of the tables and figures sometimes appears in bold and sometimes not.

Section titles are sometimes in one format and sometimes in another.

Between one section and another, there is sometimes a single space and other times up to 5 lines of space.

After each reference, there is a space between it and the endpoint of the sentence.

There are sentences merged in the absence of a full stop.

There are also merged and duplicated words, and even misspelled ones (stidy, rktes, thered...).

The common name of the species (red kite) continues to be misspelled in several places in the text.

The full name of SEO is indicated, but not that of GREFA.

The cardinal points in English are written with the first letter capitalized.

If Teflon is the commercial name, the symbol (R) must be added, if it is being used as a common name, it must be lowercase.

The correct name of the statistical analyzes is Student's t-test and Fisher's exact test.

The p-value symbol is a lowercase and italicized p.

Tables do not indicate what the abbreviations mean (SE, df, mig, sed...).

Finally it is mentioned several times as a result of "about one-fith of the individuals...". This is a scientific text, please indicate the exact percentage.

Please review carefully the manuscript and correct all this issues. 
